# Learning cross-regional dependence of EEG with convolutional neural networks for emotion classification

**Seong-Eun Moon, Soobeom Jang & Jong-Seok Lee**
School of Integrated Technology
Yonsei University
Republic of Korea
{se.moon,soobeom.jang,jong-seok.lee}@yonsei.ac.kr

## Abstract

Electroencephalography (EEG) has received much attention because it provides comprehensive information of human perception and is relatively space- and cost-effective compared with the other functional brain imaging methods. In this paper, we present an approach to model undirected and directed cross-regional dependence of EEG signals with convolutional neural networks (CNNs). It considers the brain connectivity that measures simultaneous activation of different brain regions. Furthermore, the spatial arrangement of EEG electrodes is examined. We verify the effectiveness of the method for EEG-based emotion recognition.

## 1 Introduction

Cerebral signals such as electroencephalography (EEG) that are expected to contain the comprehensive information of human perception have received much attention for human-centric services and human-computer interactions in recent years. Deep learning approaches are also often employed for such applications. For example, Zheng & Lu (2015) and Yanagimoto & Sugimoto (2016) used deep neural networks using EEG signals for emotion recognition. However, the brain connectivity that measures the relationship between different brain regions has not been considered in previous deep learning studies, although it has been shown in the neuroscience field that the cross-regional relationship is an important clue of brain functions (Friston, 2011).

A further limitation of the previous studies is that the spatial arrangement of the EEG electrodes that provides the spatial patterns of EEG signals has not been investigated well. However, the effectiveness of the spatial information of EEG signals is shown in Bashivan et al. (2016), where representing EEG power spectral density (PSD) as a two-dimensional image can enhance the performance of mental load classification.

In this paper, we present a deep learning approach to incorporate the cross-regional relationship and the spatial information of the EEG electrodes within convolutional neural networks (CNNs). We exploit both undirected and directed brain connectivity features of EEG signals with the CNNs to learn the information appearing across different brain regions through the filters in the convolutional layers.

## 2 Method

### 2.1 Connectivity features

We consider four connectivity features, namely, Pearson correlation coefficient (PCC), phase locking value (PLV) (Lachaux et al., 1999), phase lag index (PLI) (Stam et al., 2007), and transfer entropy (TE) (Schreiber, 2000). PCC measures the linear relationship between two signals and is calculated as $PCC = \frac{cov(x,y)}{\sigma_x \sigma_y}$, where $cov(\cdot)$ is the covariance, and $\sigma_x$ and $\sigma_y$ are the standard deviations of two signals $x$ and $y$, respectively. PLV accounts for the phase synchronization between two time

series, which is defined as $PLV = \frac{1}{N} \left| \sum_{n=1}^{N} e^{j \Delta \phi_n} \right|$, where $N$ is the number of time windows and $\Delta \phi_n$ is the phase difference for the $n$-th window. PLI also measures phase synchronization, but uses only the sign of the phase difference for robustness against the common source problem that is typically induced by the volume conductance effect or an active reference of the EEG signals. It is defined as $PLI = \frac{1}{N} \left| \sum_{n=1}^{N} sign(\Delta \phi_n) \right|$, where $sign(\cdot)$ indicates the sign function. TE represents the directed information flow, that is, how much the uncertainty of future prediction of a time series is reduced by knowing the past of the other time series, which is computed by $TE_{y \to x} = \sum_{n=1}^{N} p(x_{n+1}, x_n, y_n) \log_2 \frac{p(x_{n+1}|x_n, y_n)}{p(x_{n+1}|x_n)}$.

## 2.2 CONNECTIVITY MATRIX

In order to use a set of connectivity features as an input of CNNs, we need to determine how to represent them as a two-dimensional matrix. We form a connectivity matrix from the connectivity features, whose $(i, j)$-th element represents the connectivity feature between electrodes $i$ and $j$. Note that, since PCC, PLV, and PLI have no directionality, the connectivity matrices for these are symmetric, unlike TE. The order of the electrodes in the connectivity matrix may become important because filters in a CNN model the patterns appearing in adjacent values in the connectivity matrix.

We suggest an ordering method considering the spatial arrangement of the EEG electrodes, that is, the electrodes are arranged depending on the physical distance between them. This is because the EEG signals of two physically adjacent electrodes are usually correlated due to the volume conductance effect. Specifically, the order of electrodes starts from the left frontal electrode, and continues to the closest electrode. In addition, we also consider a randomly determined order to comparatively examine the effect of ordering.

## 3 CASE STUDY: EMOTION RECOGNITION

### 3.1 EXPERIMENTAL SETUP

The Database for Emotion Analysis using Physiological signals (DEAP) (Koelstra et al., 2012) is employed to verify the effectiveness of our method, which is one of the largest databases and has been popularly used for EEG-based emotion classification. The database includes one-minute-long EEG signals of 32 subjects measured during the watching of 40 affective music videos. The EEG signals were recorded using a 32-channel EEG recording system.

The problem of binary classification of valence is considered, where a given set of EEG signals is classified whether the induced emotion is positive or negative. The subjective rating has a 9-point rating scale, so we define the scores ranging from 1 to 5 as the low valence class, and those from 6 to 9 as the high valence class. The former takes 55.31% of the entire data, and 44.69% of the data correspond to the latter.

The EEG signals are divided into three-second-long segments with an overlap of 2.5 seconds. Thus, we obtain 115 segments of EEG signals per video. The segmented signals for all videos are split into five clusters randomly, so that a five-fold leave-one-cluster-out cross-validation scheme is implemented. To obtain signal components for ten frequency bands, bandpass filtering is applied for delta (0-3 Hz), theta (4-7 Hz), low alpha (8-9.5 Hz), high alpha (10.5-12 Hz), alpha (8-12 Hz), low beta (13-16 Hz), mid beta (17-20 Hz), high beta (21-29 Hz), beta (13-29 Hz), and gamma (30-50 Hz).

The connectivity features are extracted from EEG signals for every pair of electrodes. For comparison, we also employ PSD, which is commonly used to describe the activation level of an EEG signal. It is obtained by using the Welch's method and transformed to a 32×32 topography based on the location information of the EEG electrodes.

We design three different CNN structures. The first one is the simplest, having a convolutional layer and a max-pooling layer (denoted as Net1). The second CNN has three convolutional layers and two max-pooling layers (Net2), i.e., two convolutional layers and one max-pooling layer are added after the max-pooling layer in Net1. The most complex CNN has five convolutional layers and five max-pooling layers, one after the other (Net3). We also tried more complex structures, but did not

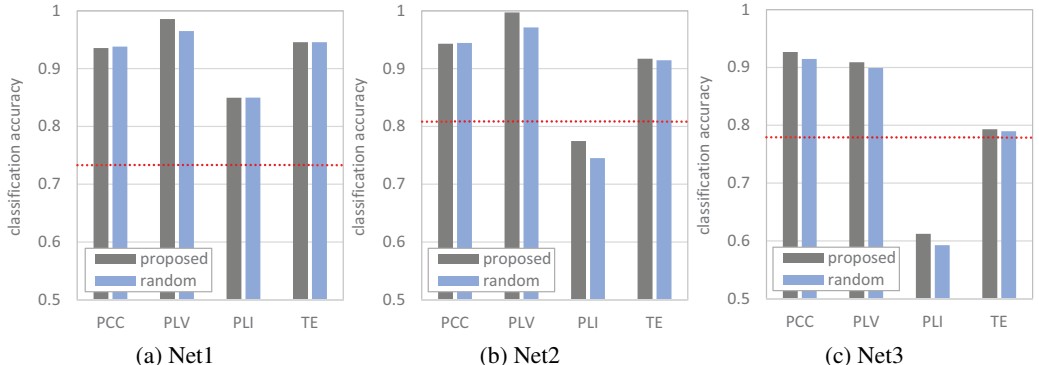

Figure 1: Classification accuracy for different network structures and EEG features. Dotted lines indicate the classification accuracy obtained with PSD features.

observe improved classification performance. A fully connected layer with 256 hidden nodes is added after the last max-pooling layer in all CNN structures to obtain the output defined as 0 for the low valence class and 1 for the high valence class. The size of all filters in the convolutional layers is set to 3×3. The number of filters is 32 in the first layer and then becomes twice that of the previous convolutional layer. We use the ReLU activation function for the convolutional layers. The batch normalization is used after max-pooling.

We implement the CNNs in Theano. The Adam algorithm is used to minimize the loss in terms of the cross-entropy function. The batch size is set to 256.

## 3.2 Results

The average classification accuracy from the five-fold cross-validation is shown in Figure 1. Overall, the results are better than that obtained by using a support vector machine (SVM) classifier having the radial basis function kernel (0.554). This shows that CNNs can improve the performance of emotion recognition compared with the conventional SVM, which is because the spatial information of EEG signals is not ignored in CNNs.

We obtain the best result (0.997) for Net2 with the PLV matrices formed with the proposed ordering method. This is a significant improvement over the performance of the PSD features, i.e., 0.809 obtained using Net2. Moreover, the connectivity features always yield better performance than PSD except for PLI. That is, our method for utilizing connectivity features effectively enhances the performance.

The two ordering methods show comparable results in most cases, however, our ordering method shows noticeable improvement in some cases. Particularly, the best performance is obtained by the method considering spatial information, showing a relative error rate reduction of 89.7% (0.971 to 0.997) in comparison to the random ordering method. Therefore, it is better to let physically adjacent electrodes be located in neighboring columns and rows in a connectivity matrix so that CNNs learn their relationship easily.

Among the connectivity features, PLV is the best for all cases except for Net3, where PCC outperforms PLV. When the two phase-related features (i.e., PLV and PLI) are compared, PLV shows higher accuracies than PLI. The sign function used in PLI seems to remove useful information. TE does not show any improvement compared to the other connectivity features despite of consideration of information directionality. The way to consider directionality may not be effective in TE. Thus, future work on developing PLV-like features with directionality could be interesting.

## Acknowledgments

This work was supported by Basic Science Research Program through the National Research Foundation of Korea (NRF) funded by the Korea government (MSIT) (NRF-2016R1E1A1A01943283).

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
