# OpenReview forum: "Learning cross-regional dependence of EEG with convolutional neural networks for emotion classification"
_ICLR.cc/2018/Workshop — Reject_

### Official Review · AnonReviewer2 · 2018-03-10
**Deep networks for emotion classification from EEG signals**

**Rating:** 3
**Confidence:** 5

**Review:**

The paper studies the usage of deep networks for binary classifying emotions on the basis of various (connectivity) features extracted from EEG of subjects watching videos. Essentially it finds that the neural network does better than SVMs, although I did not find information on the SVM part.
The paper is very preliminary. At the current point it should not be accepted. There are a number of issues:

- it is completely unclear what the neural network does to achieve good classification rates. There should be meaningful physiological understanding, what the network has learned, e.g. whether these are artifacts or other things
- the 5-fold crossvalidation is not meaningful as there are strong dependencies in the EEG data. Leave one subject out is more meaningful or blockwise cross-validation schemes
- no proper comparison to linear baselines is given, also the SVM part is not documented
- it is unclear what are the effects of volume conduction
- it is unclear, which connectivity measures work well for the network

---

### Official Review · AnonReviewer3 · 2018-03-12
**Review of learning cross-regional dependence of EEG with convolutional neural networks for emotion classification**

**Rating:** 8
**Confidence:** 3

**Review:**

This paper proposes to using connectivity features as input of deep learning architectures. The authors evaluate different features, namely Pearson coefficient, phase locking value, phase lag index and transfer index, to build connectivity matrices that are fed to a convolutional architecture of different depths. The experiments are conducted on DEAP dataset and shows that a 5-layers net yields the best results. It could be interesting to see how this scheme is robust to interindividual variation: instead of using a leave-one-cluster-out, the authors could use a leave-one-subject out. This scheme avoid the contamination of subject-specific information in training dataset.

---

### Official Review · AnonReviewer1 · 2018-03-13
**Is there overlapping  data in the training and test sets?**

**Rating:** 3
**Confidence:** 4

**Review:**

While the idea of using connectivity measures with deep learning for classification is
worth exploring, it is not strictly speaking the first combination
of deep learning and connectivity measures (see for example
https://github.com/alexandrebarachant/Grasp-and-lift-EEG-challenge ).
However comparing the different connectivity measures is novel and worth
exploring.

However, I have one major concern with this paper -- given that
segments are overlapping, do you make sure that overlapping segments
are not in the same cross-validation fold?  Given that the filtering
will also blur information over time, the different folds should even
have a time gap between them so that there are no segments closer than
a certain time interval between the training and test sets.  With
overlapping segments in the training and test dataset, it is possible
that the high capacity networks you are using are performing similar
to a nearest neighbor classify and classifying overlapping segments
with the same label -- This means that the classifier does not
necessarily classify based on emotion, but is effectively just
classifying time.

The concerning text is on page 2 and reads

"The EEG signals are divided into three-second-long segments with an
overlap of 2.5 seconds. Thus, we obtain 115 segments of EEG signals
per video. The segmented signals for all videos are split into five
clusters randomly, so that a five-fold leave-one-cluster-out
cross-validation scheme is implemented."

I am interpreting this to mean that segments from one video were
randomly sorted into the five clusters allowing overlapping segments
to appear in training and test sets.  If this is an incorrect
interpretation, then I would be	in favor of acceptance of this paper.

The original paper
https://www.eecs.qmul.ac.uk/mmv/datasets/deap/doc/tac_special_issue_2011.pdf
that introduced the dataset (and has the senior author of this paper  as a co-author)
was careful to state "At each step of the cross validation, one video was used as the
test-set and the rest were used as training-set."

---

### Decision · Program_Chairs · 2018-03-20
**ICLR 2018 Workshop Acceptance Decision**

**Decision:**

Reject

**Comment:**

Based on the reviews, this paper has not been accepted for presentation at the ICLR workshop. However, the conversation and updates can continue to appear here on OpenReview. As discussed by the reviewers, when encouraging ICLR to study applications such as EEG more, it is important to introduce good experimental practice.